# SPaRFT: Self-Paced Reinforcement Fine-Tuning for Large Language Models

## Abstract

Large language models (LLMs) have shown strong reasoning capabilities when fine-tuned with reinforcement learning (RL). However, such methods require extensive data and compute, making them impractical for smaller models. Current approaches to curriculum learning or data selection are largely heuristic-driven or demand extensive computational resources, limiting their scalability and generalizability. We propose **SPaRFT**, a self-paced learning framework that enables efficient learning based on the capability of the model being trained through optimizing which data to use and when. First, we apply *cluster-based data reduction* to partition training data by semantics and difficulty, extracting a compact yet diverse subset that reduces redundancy. Then, a *multi-armed bandit* treats data clusters as arms, optimized to allocate training samples based on model current performance. Experiments across multiple reasoning benchmarks show that SPaRFT achieves comparable or better accuracy than state-of-the-art baselines while using up to $100\times$ fewer samples. Ablation studies and analyses further highlight the importance of both data clustering and adaptive selection. Our results demonstrate that carefully curated, performance-driven training curricula can unlock strong reasoning abilities in LLMs with minimal resources.

## 1 Introduction

Large Language Models (LLMs) have achieved remarkable progress in tasks requiring reasoning, problem-solving, and generalization, driven largely by scaling trends in model size, data, and compute (Google, 2024; OpenAI, 2024). As the cost and complexity of pretraining continue to rise, research attention has increasingly shifted toward post-training techniques, which aim to improve LLM capabilities more efficiently. Among these, Reinforcement Fine-Tuning (RFT) has emerged as a promising method that aligns model behavior with outcome-based reward signals, often relying on lightweight supervision without elaborate reward engineering or inference-time computation (Kumar et al., 2025; DeepSeek-AI, 2025; Lightman et al., 2023).

Standard RFT trains on uniformly sampled batches from the full dataset (DeepSeek-AI, 2025). While simple, this approach ignores each example's difficulty, informativeness, and uncertainty, wasting limited reward feedback on trivial or noisy instances and slowing convergence (Ouyang et al., 2022; Dong et al., 2023). This raises two key underexplored dimensions: how to select which examples to train on, and how to present them to LLMs over training.

Data reduction methods prioritize informativeness by estimating example difficulty or uncertainty. For example, variance-based filtering based on multiple forward passes through a reference model (Wang et al., 2025b). Although effective at denoising, these approaches incur significant computational overhead, making them impractical for resource-constrained models. They are also sensitive to both the selected training examples and the uncertainty estimator, which can hinder generalization to new LLMs.

In parallel, curriculum design plays a central role in guiding the learning trajectory (Bengio et al., 2009). As the model improves during fine-tuning, the useful difficulty level shifts dynamically, yet static curricula or random orders often fail to reflect this progression. Recent attempts at adaptivity filter examples with heuristic thresholds (Shi et al., 2025), but such mechanisms can be fragile, especially for small or weak models that rely on imperfect difficulty metrics, prematurely excluding challenging, informative examples early and stalling progress.

We introduce **SPaRFT**, a self-paced RFT framework that automatically selects informative training examples and designs an adaptive schedule to improve LLM training efficiency. We formulate RFT as a Multi-Armed Bandit (MAB) (Sutton et al., 1998), where each arm is a cluster with similar semantics and difficulty. The MAB decides which data to present at each step, overcoming rigid, heuristic-driven assignment. First, we reduce data to form the clusters (arms), clustering on latent representations and per-example attributes. Motivated by redundancy in training data for RFT (Wang et al., 2025b), we retain a fixed number of samples per cluster; within each, representative examples are chosen by iteratively maximizing embedding distance from previously selected samples. In the second stage, we optimize cluster selection based on training performance: pulling an arm equals to sampling from that cluster, and the inverse of the current solve rate serves as the reward to update the bandit. Our intuition is simple: *prioritize what is currently challenging*. This avoids retraining on already-solved instances—crucial in low-resource settings with tiny LLMs and tight budgets (Le et al., 2025)—where curricula and reduction must be efficient, performance-aware, and free of costly heuristics to make RFT practical in the small-model regime.

To evaluate our approach, we conduct extensive experiments on mathematical problem-solving tasks using various LLMs of different sizes. Results show that SPaRFT significantly improves reasoning accuracy and robustness compared to both reinforcement learning and curriculum learning baselines, while reducing the number of training examples by a factor of 100. Notably, SPaRFT outperforms methods that rely on exhaustive search to select single or few training examples. Our analysis reveals how poorly designed curricula can get stuck in easy example regions, failing to leverage the diversity of the dataset.

In summary, our contributions are threefold: (1) We propose **SPaRFT**, a novel two-stage framework that reduces the number of training examples and optimizes the RFT progress for LLMs using MAB. (2) Our method is lightweight, significantly reducing the number of training examples while adding minimal computational overhead, making it well-suited for RFT of tiny LLMs. (3) Our extensive experiments demonstrate that SPaRFT consistently outperforms existing curriculum and data reduction strategies.

## 2 RELATED WORK

### 2.1 DATA REDUCTION FOR REINFORCEMENT FINE-TUNING OF LANGUAGE MODELS.

The role of data in RFT for LLMs remains an open area of research. Several works attempt to curate high-quality mathematical datasets (Luo et al., 2025; Yu et al., 2025), but they do not explicitly explore which data is most effective for fine-tuning. More recently, alternative approaches have proposed using heuristic-based scores, such as Learning Impact Measurement (Li et al., 2025) or variance-based selection methods (Wang et al., 2025b). These techniques alleviate data constraints by enabling training on only a small subset while still achieving strong reasoning capabilities. However, these methods typically require significant pre-computation, limiting their practicality in real-world deployment, and remain untested on small models with limited reasoning capabilities.

### 2.2 CURRICULUM LEARNING FOR LLMS.

Humans and animals learn more effectively when examples are presented in a meaningful order that gradually increases in complexity. Curriculum learning (Bengio et al., 2009) and performance-guided training progression (Le et al., 2022) have been applied to supervised and RL training. For LLMs, recent studies have explored how to organize training data to reduce computational cost and improve sample efficiency, though this area remains underdeveloped. Existing approaches include hand-crafted difficulty tiers (Wen et al., 2025; Luo et al., 2025; Song et al., 2025), which often require task-specific insights and manual tuning. More adaptive methods, such as AdaRFT (Shi et al., 2025), learn a training curriculum by dynamically adjusting a difficulty threshold to select examples. While promising, these methods still face limitations: repeated training on easy examples can lead to overfitting or poor generalization; difficulty heuristics may not transfer across tasks; and fixed sampling strategies may fail to adapt to evolving model capabilities. There remains a need for more principled curriculum strategies tailored to the scale and dynamics of LLM training.

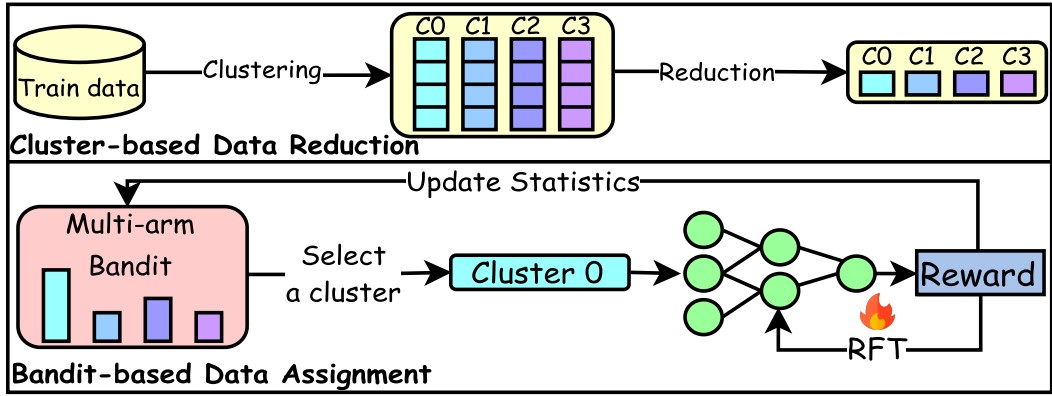

Figure 1: SPaRFT Architecture. Top: Initial training data is annotated with difficulty, and each example's feature vector is formed by concatenating its latent embedding with its difficulty score. These vectors are clustered into $K$ groups, and from each cluster we select representative samples to ensure both coverage and diversity. Bottom: Each cluster is treated as an arm in a multi-armed bandit. At each step, Thompson Sampling is used to select a cluster, its representative examples are fed to the LLM to obtain rewards for RFT, and those rewards are used to update the bandit statistics.

# 3 SELF-PACED REINFORCEMENT FINE-TUNING

We aim to improve a policy $\pi_\theta$ by adaptively presenting training samples while minimizing the required data. Focusing on tasks that are too easy or too hard is inefficient, offering little challenge or feedback. Instead, data assignment should adapt to the model's evolving capabilities, presenting examples that it is ready to learn at each stage. Threshold-based curricula require manual tuning and can be suboptimal, especially at the early training stage (Shi et al., 2025). To address this, we introduce **SPaRFT**, a two-phase approach for self-paced optimization: (1) cluster-based data reduction and (2) bandit-based data assignment. SPaRFT integrates with common RFT algorithms; we use GRPO (DeepSeek-AI, 2025) by default. A detailed description appears in Algorithm 1, Appendix A.1.

## 3.1 CLUSTER-BASED DATA REDUCTION

### 3.1.1 DATA CLUSTERING

RFT of large language models often presupposes access to abundant, high-quality supervision, which is costly or impractical in low-resource settings. We introduce a clustering-based data reduction procedure that lowers data requirements while preserving or improving training efficacy. The approach leverages two signals per training instance: (i) a latent representation and (ii) a scalar, per-example attribute available from the dataset. Grouping examples that are proximate in latent space and exhibit comparable attribute values yields a coherent partitioning that is well suited to curriculum-style sampling.

*Latent representation.* For each example, we obtain its latent representation with a pre-trained embedding model. To mitigate the well-known degradation of distance metrics in high dimensions (Bellman, 1966), we apply Principal Component Analysis (PCA) to reduce dimensionality. For notation, let $x_i$ denote the $i^{th}$ training example and $s_i$ denotes its PCA-reduced latent vector.

*Per-example attribute.* Our framework supports the inclusion of an optional scalar attribute at the per-example level, alongside semantic embeddings. Let $d_i$ denote this attribute, which can flexibly encode any task-specific signal. For clustering, we concatenate $d_i$ with the latent embedding to form a joint representation. This design enables clustering to account not only for semantic similarity but also for structural or pedagogical cues that are important for curriculum construction.

*Clustering.* Prior to clustering, we standardize the coordinates of $s_i$ and $d_i$ to zero mean and unit variance, preventing either modality from dominating the distance metric. We then form the com-

bined representation $e_i$ as follows:

$$e_i = \hat{s}_i \oplus \hat{d}_i, \tag{1}$$

where $\hat{s}_i$ and $\hat{d}_i$ are the standardized latent vector and scalar feature, respectively, and $\oplus$ denotes concatenation. Finally, we apply $k$-means clustering to $\{e_i\}$ to partition the dataset into $K$ clusters (Lloyd, 1982), thereby ensuring coverage across the joint latent–attribute space.

### 3.1.2 DATA REDUCTION

Recent work indicates that, in the context of scaling RFT, strategically curated subsets of training data can yield stronger LLM reasoning performance than finetuning on the full corpus (Li et al., 2025; Wang et al., 2025b). Motivated by this, we first partition the training set into $K$ clusters as mentioned above, and then subsample a fixed quota of representatives per cluster, chosen to preserve coverage and diversity. Let $l$ denote the number of examples for each cluster, $C_k$ denote the set of examples in cluster $k^{th}$, and let $\mu_k$ be the corresponding cluster centroid in the embedding space. For each embedded representation $e_i$ in cluster $k$, we compute its Euclidean distance to the its cluster centroid:

$$\delta_i = \|e_i - \mu_k\|_2, \tag{2}$$

We then sort all examples in $C_k$ by their distances $\delta_i$. To construct a diverse and representative training subset from each cluster, we employ a greedy farthest-point sampling strategy: starting from the cluster centroid, we iteratively select examples that are maximally distant from those already chosen. This balances centrality and coverage in the embedding space. The full procedure is outlined in Phase 1 of Algorithm 1 in Appendix A.1.

## 3.2 BANDIT-BASED DATA ASSIGNMENT

Our core intuition is to prioritize examples that are *currently* challenging for the model rather than fixing a static "hard set", since what is difficult changes as training progresses. At each step $t$, the policy $\pi_\theta$ selects a cluster $c_t \in \{1, \ldots, K\}$, framing scheduling as a multi-armed bandit where each arm is a cluster. We use Thompson Sampling (Thompson, 1933; Russo et al., 2020): for cluster $k$, maintain cumulative reward $R_k^{(t)}$ and count $n_k^{(t)}$, and draw as follows:

$$\tilde{\mu}_k^{(t)} \sim \mathcal{N}\left(-\frac{R_k^{(t)}}{n_k^{(t)} + \epsilon}, \frac{1}{n_k^{(t)} + \epsilon}\right). \tag{3}$$

Here, the mean is negated so clusters with lower *solve rates* (harder under the current model) are favored. We especially note that solve rate is computed online and differs from the precomputed *difficulty* $d_i$ used only for clustering. The selected cluster as follows:

$$c_t = \arg\max_k \tilde{\mu}_k^{(t)}. \tag{4}$$

This formulation naturally balances exploration and exploitation: clusters with fewer observations are more likely to be explored, while clusters with consistently lower solve rates (i.e., harder samples) are exploited more frequently. A batch of size $B$ is sampled from cluster $c_t$ to train $\pi_\theta$. The reward for each sample in the batch is computed based on the correctness of the model's output:

$$r_i = \begin{cases} 1, & \text{if the response is correct} \\ 0, & \text{otherwise} \end{cases} \tag{5}$$

The average reward over the batch is calculated as $r_{\text{avg}} = \frac{1}{B}\sum_{i=1}^{B} r_i$. This average reward is used in two ways. First, it provides a scalar learning signal to update the policy network $\pi_\theta$ via any RL algorithm (e.g., GRPO). The RL training objective is to maximize the expected total reward:

$$\max_\theta \mathbb{E}_{q \sim D_{train}, a \sim \pi_\theta}[r_{avg}] \tag{6}$$

where $a$ is the sampled answer from $\pi_\theta$ given the question $q$.

Second, $r_{avg}$ is used to update the internal statistics of the multi-armed bandit by adjusting both the cumulative reward and the count of interactions for the selected cluster:

$$R_{c_t}^{(t+1)} = R_{c_t}^{(t)} + r_{\text{avg}}, \quad n_{c_t}^{(t+1)} = n_{c_t}^{(t)} + 1. \tag{7}$$

This dynamic update process ensures that the sampling distribution adapts online to the evolving state of the model. Clusters that temporarily present higher error rates receive increased sampling probability, while those that become easy are sampled less frequently. Crucially, because rewards are derived from model predictions themselves, this approach tightly couples curriculum scheduling with real-time model capacity, which results in a self-regulating training signal that continuously steers focus toward maximally informative examples. Full algorithm of SPaRFT is provided in Algorithm 1, Appendix A.1. In addition, we formally analyze the convergence of the MAB used in SPaRFT, as stated in the following Proposition.

**Proposition 1.** *Under assumptions: (i) bounded rewards, LLM training with (ii) gradient clipping and (iii) decayed learning rate, the Thompson Sampling scheduler in SPaRFT satisfies sublinear reward variation up to step $T$: $V_T = O(\log T)$. Consequently, as $t \to \infty$, the sampling distribution concentrates on clusters with maximal expected reward.*

*Proof.* See Appendix A.2. □

## 4 EXPERIMENTS

We evaluate our method on several LLMs. Full fine-tuning is performed on *Qwen3-0.6B*, *Qwen2.5-0.5B-Instruct*, *Falcon3-1B-Instruct*, and *Llama3.2-1B-Instruct* using a single NVIDIA H100 GPU, with *Qwen3-0.6B* achieving performance comparable to larger models (e.g., *Qwen2.5-Math-7B-Instruct*). In addition, we fine-tune *Qwen3-8B-Base* with Low-Rank Adaptation (Hu et al., 2021) on a single H200 GPU. Training covers mathematical and logical reasoning datasets: (1) DeepScaleR-Uniform, (2) DeepScaleR-Easy, (3) DeepScaleR-Difficult (10k problems each) (Luo et al., 2025); (4) GSM8K (7,473

| Method | GSM8K | MATH500 | AIME24 | AIME25 |
|--------|-------|---------|--------|--------|
| Base | 78.0 | 75.4 | 10.0 | 16.7 |
| $\pi_1$ | 78.1 | 75.4 | 13.3 | 16.7 |
| $\pi_2$ | 79.0 | 74.2 | 6.7 | 10.0 |
| Ordered | 77.9 | 74.8 | 13.3 | 13.3 |
| SFT | $77.6_{0.5}$ | $74.8_{0.4}$ | $7.8_{1.9}$ | $14.4_{5.1}$ |
| R1 | $77.9_{1.0}$ | $71.6_{1.3}$ | $8.9_{5.1}$ | $12.2_{5.1}$ |
| AdaRFT | $78.9_{0.3}$ | $75.9_{1.5}$ | $12.2_{3.9}$ | $14.4_{3.9}$ |
| **SPaRFT** | $\mathbf{79.5}_{0.6}$ | $\mathbf{78.0}_{1.3}$ | $\mathbf{14.4}_{1.9}$ | $\mathbf{18.9}_{5.1}$ |

Table 1: Results using Qwen3-0.6B across different datasets. We report extractive match scores ($\text{mean}_{\text{std}}$) at the final training checkpoint, averaged over 3 seeds (except for the Base, $\pi_1$, $\pi_2$, and Ordered baselines). Best results are highlighted in bold, and the second-best are underlined.

problems) (Cobbe et al., 2021); and (5) Knights and Knaves (K&K) (Xie et al., 2024). Latent embeddings for clustering are obtained using *Qwen3-Embedding-0.6B*. All chosen datasets are annotated with difficulty levels, either instantiated as a per-example attribute defined by solve rates from a moderate LLM (Shi et al., 2025) or explicit labels (Xie et al., 2024). While our framework can use other attributes (e.g., uncertainty, informativeness), **we choose difficulty as the per-example attribute** for clustering since it is natural and interpretable. Each experiment is repeated with three random seeds, and training is implemented with the Open-R1 codebase (Hugging Face, 2025). In each run, our method selects **only up to 100 training examples** using the reduction strategy in Section 3. This process adds negligible overhead compared to R1: on *Qwen3-0.6B* with an H100 GPU, training time is nearly identical (15h37m vs. 15h23m), with only a slight increase due to bandit-based selection.

**Evaluation**  We use five benchmarks that span different reasoning types and difficulty levels: **GSM8K** (Cobbe et al., 2021), consists of diverse grade school math problems; **MATH500**, a 500-sample subset of the MATH dataset (Hendrycks et al., 2021); **AIME24** and **AIME25**, comprising problems from the 2024 and 2025 American Invitational Mathematics Examination, respectively; and finally, the logical reasoning **K&K** test set consists of 700 samples, with 100 examples for different number of people in the question from 2 to 8. We report the extractive match scores for all mathematical datasets, following Lighteval's evaluation framework (Habib et al., 2023). For K&K dataset, we follow the evaluation protocols established by the dataset authors Xie et al. (2024).

**Baselines**  **Base** refers to the pretrained model without any fine-tuning. $\pi_1$ and $\pi_2$ represent the baselines in **1-shot RLVR** paper, trained on one and two examples selected from the DeepScaleR

| | Number of People | | | | | | | Average |
|---|---|---|---|---|---|---|---|---|
| Method | 2 | 3 | 4 | 5 | 6 | 7 | 8 | |
| Base | 32.0 | 10.0 | 8.0 | 2.0 | 0.0 | 0.0 | 0.0 | 7.4 |
| R1 | 31.7±1.5 | 11.3±0.6 | 8.3±0.6 | 4.0±1.0 | 0.7±0.6 | 0.0±0.0 | 1.0±0.0 | 8.1±0.3 |
| ADARFT | 31.7±0.6 | 10.7±2.1 | 7.3±0.6 | 3.7±0.6 | 0.3±0.6 | 0.0±0.0 | 1.0±0.0 | 7.8±0.2 |
| **SPaRFT** | **34.3±1.2** | **13.7±1.2** | **10.0±1.0** | **5.3±1.5** | **1.6±1.5** | 0.0±0.0 | **1.3±0.6** | **9.5 ±0.8** |

Table 2: Accuracy (%) by number of people in K&K puzzles with results reported as mean ± standard deviation (except for Base baseline) over 3 runs. Bold denotes the best mean performance.

dataset, respectively (Wang et al., 2025b). **SFT** denotes the supervised fine-tuning baseline. **Ordered** (Bengio et al., 2009) is a curriculum baseline in which training begins with easier examples and gradually progresses to harder ones. **R1** is the RL baseline trained with the standard GRPO algorithm without an SFT cold start, as in DeepSeek-R1 (DeepSeek-AI, 2025). **AdaRFT** is a curriculum learning approach that selects examples based on a difficulty threshold (Shi et al., 2025). We also include an additional variance-based baseline, **LIMR** (Li et al., 2025), trained on the MATH dataset (Hendrycks et al., 2021), with details in Appendix A.4.

## 5 EXPERIMENTAL RESULTS

### 5.1 BENCHMARKING SPARFT WITH QWEN3-0.6B

We fine-tune *Qwen3-0.6B* in a zero-shot setup. Despite its small size, it is a strong baseline and supports a *thinking mode* using `<think> </think>` tags. Following (Wang et al., 2025b), we use one seed for $\pi_1$ and $\pi_2$. For the *Ordered* baseline, we also use one seed, as the order of training examples is fixed. Further details on training hyperparameters are provided in Appendix A.13.

Table 1 presents test accuracies from the final training checkpoint using *Qwen3-0.6B*. SPaRFT consistently achieves the highest performance across all benchmarks, despite using only 100 training examples—*two orders of magnitude fewer than other baselines*. On GSM8K, SPaRFT reaches $79.5 \pm 0.6\%$, outperforming all baselines including AdaRFT ($78.9 \pm 0.3\%$) and R1 ($77.9 \pm 1.0\%$). On the MATH500 benchmark, SPaRFT yields $78.0 \pm 1.3\%$, improving over the next-best result ($75.9 \pm 1.5\%$) by a margin of 2.1 percentage points. Notably, SPaRFT also shows strong gains in harder math datasets: it reaches $18.9 \pm 5.1\%$ on AIME25, outperforming AdaRFT ($14.4 \pm 3.9\%$) by 4.5 percentage points, and achieves the best score on AIME24 with $14.4 \pm 1.9\%$. These results highlight SPaRFT's data efficiency and robustness across tasks of varying difficulty.

### 5.2 SPARFT WORKS WITH VARIOUS DATASETS

#### 5.2.1 MATHEMATICAL DATASETS TRAINING RESULTS

We evaluate our method on three distinct training sets: (1) GSM8K; (2) DeepScaleR–Easy, a subset of DeepScaleR with primarily low-difficulty questions; and (3) DeepScaleR–Difficult, a subset with mainly high-difficulty questions. All experiments use *Qwen3-0.6B* as the backbone, and we adopt the same reward functions and hyperparameters as in Section 5.1. We compare with the top-3 baselines, excluding $\pi_1$ and $\pi_2$ as they are unavailable in GSM8K. As seen in Figure 2, across all settings, SPaRFT consistently outperforms standard SFT, achieving gains of 2.9–5.5 % on GSM8K and up to 7.8 % on the AIME benchmarks. R1 generally ranks second, especially on the easier splits, while AdaRFT falls 1–2 % behind in most cases. Notably, when trained on the difficult subset (3), SPaRFT attains a 2.3 % improvement on MATH500 and more than doubles AIME24 accuracy relative to SFT. These results confirm that SPaRFT not only enhances overall accuracy but also yields the greatest benefits on the most challenging training set.

#### 5.2.2 K&K TRAINING RESULTS

In this dataset, we consider the number of people in the question as the per-example attribute $d_i$ and select *Qwen3-0.6B* as the Base LLM for the experiments. The final answer is used for computing the

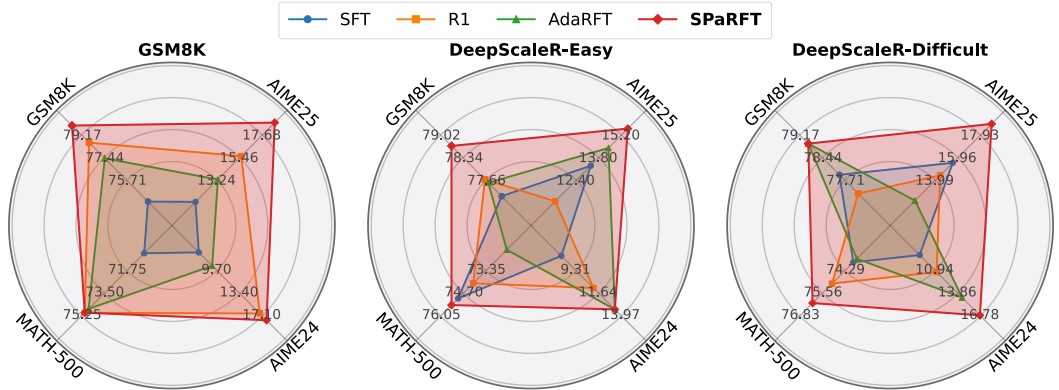

Figure 2: Results averaged over 3 training seeds using three training datasets with Qwen3-0.6B.

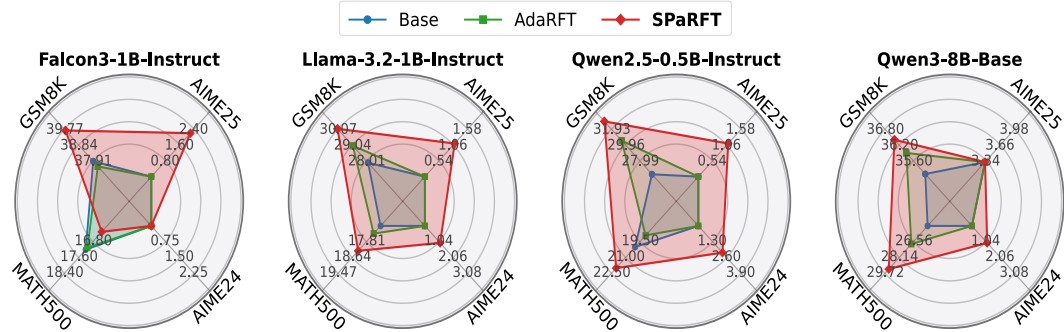

Figure 3: Results averaged over 3 training seeds using other LLMs across datasets.

accuracy reward, and the evaluation protocol strictly follows the original setup in Xie et al. (2024) to ensure consistency and comparability. As shown in Table 2, SPaRFT consistently outperforms all baselines across difficulty levels, with clear gains on more complex puzzles involving 2–5 people, where reasoning demands are substantially higher. It achieves the highest overall average accuracy of 9.5%, compared to 8.1% for R1 and 7.8% for AdaRFT–representing relative improvements of ~17% and ~22%, respectively. Notably, the AdaRFT baseline shows a decline in accuracy in 2 out of 7 settings, indicating that training with a noisy curriculum can negatively impact model performance. These results collectively highlight the effectiveness of SPaRFT in scaling to harder reasoning cases while maintaining competitive performance on simpler ones, thereby validating the role of curriculum-guided selection in enhancing reasoning-focused training.

### 5.3 SPARFT HELPS DIVERSE LLM LEARNERS

We train on DeepScaleR-Uniform and evaluate *Qwen2.5-0.5B-Instruct*, *Falcon3-1B-Instruct*, *Llama3.2-1B-Instruct*, and *Qwen3-8B-Base*, which are compact to mid-size LLMs with strong reasoning, language, code, and math skills. We exclude *Qwen3-8B* (reasoning-enabled) due to the substantial compute from long `<think>` traces. We compare against the Base model and AdaRFT, the most consistent and second-best method in Section 5.1. All models use the same zero-shot setup, except *Llama3.2-1B-Instruct*, which requires one in-context example per instance to yield valid correctness rewards (Le et al., 2025). Figure 3 shows SPaRFT as the clear winner, ranking first in 13/16 cases. Average gains are evident on GSM8K (~+3%) and MATH500 (~+2%). On the harder AIME splits, SPaRFT turns near-zero baseline scores into consistent positives, reflecting better sample efficiency under sparse-reward RFT. We see complete or near-complete sweeps on the smaller models over Base and AdaRFT, indicating benefits in capacity-constrained settings; results on *Qwen3-8B-Base* remain strong despite not being used for RFT training. Overall, a bandit-driven, performance-aware curriculum generalizes across architectures and tasks with minimal protocol changes, delivering reliable gains under compute-conscious budgets.

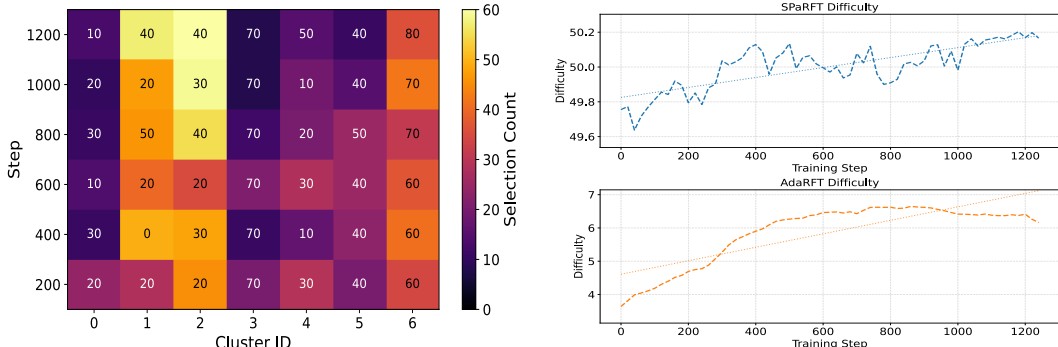

Figure 4: **Left:** Multi-arm Bandit Cluster Selection and its Impact on Cluster Solve Rates During Training. **Right:** Top: Difficulty of training examples selected by SPaRFT over time. Bottom: Difficulty of training examples selected by AdaRFT over time. Note: The plots are shown separately due to the large difference in their difficulty scales.

## 6 ABLATION STUDIES AND MODEL ANALYSES

### 6.1 MULTI-ARM BANDIT ANALYSIS

**Empirical Convergence** SPaRFT leverages the MAB framework to adaptively guide curriculum learning, making it important to characterize how the scheduler evolves during training. Figure 4 (Left) illustrates a heatmap of cluster solve rates alongside bandit selections over time using *Qwen3-0.6B* on the DeepScaleR-Uniform dataset partitioned into 7 clusters. Early in training (around step 200), the bandit behaves nearly uniformly, allocating samples across clusters with little preference. As the model accumulates experience, clear patterns emerge: clusters 1 and 2 exhibit higher solve-rate differentials, and the bandit correspondingly shifts toward sampling them more frequently, signaling that these clusters provide greater marginal learning benefit. From step 600 onward, this concentration intensifies, with clusters 1 and 2 dominating the selection distribution, indicating that the scheduler successfully adapts to focus training on regions of the data that remain most informative for continued performance improvement.

**Solve Rate Trends.** Clusters 1 and 2 show the largest solve rate gains, increasing from 20% at step 200 to 40% by step 1200. These clusters appear moderately difficult and provide strong learning signals. In contrast, cluster 3, which is rarely picked, starts high at 70% and remains flat, suggesting it is too easy to help the model improve. Other clusters, such as 4 and 5, show modest gains with continued exploration, except for cluster 0. By the end of training, the bandit converges to favor clusters 1 and 2, which consistently yield the most benefit.

### 6.2 DATA REDUCTION ANALYSIS

**Average Sample Difficulty Comparison** We consider *Qwen3-0.6B* on DeepScaleR-Uniform with 7 clusters to examine selected example difficulties. Figure 4 (Right) compares the average difficulty of training examples chosen over time by SPaRFT and AdaRFT. Although AdaRFT accesses the full dataset, its performance with tiny LLMs struggles to reach medium and hard examples,

| Method | GSM8K | MATH500 | AIME24 | AIME25 |
|---|---|---|---|---|
| SPaRFT$^-$ | 78.8±0.4 | 76.4±0.7 | 12.2±3.9 | 11.1±1.9 |
| SPaRFT | **79.5±0.6** | **78.0±1.3** | **14.4±1.9** | **18.9±5.1** |

Table 3: Mean ± standard deviation over 3 seeds on the DeepScaleR-uniform dataset using Qwen3-0.6B as the base model. SPaRFT$^-$ denotes the variant without data reduction. Best results are in bold.

due to a threshold mechanism that is highly sensitive and skews selection. In contrast, SPaRFT favors medium–hard instances, avoiding the overemphasis on easy examples seen in AdaRFT. We attribute this to our clustering strategy, which captures both semantic diversity and difficulty: each cluster mixes a broad range of examples, enabling exploration of harder cases without sacrificing

variety. This contributes to SPaRFT's superior performance over other curricula or selection methods.

**Impact of Data Reduction** To evaluate the effectiveness of the data reduction phase in our method, we conduct ablation experiments where data reduction is disabled, where all examples within each cluster are retained without filtering. Results are shown in Table 3. Without data reduction, each cluster contains significantly more examples than the batch size $B$. As shown, SPaRFT without data reduction still outperforms the Base baseline but underperforms the full version with selection. We attribute this to increased sampling variance in large, unfiltered clusters, where heterogeneity in example difficulty reduces the stability and efficacy of the learning signal.

## 6.3 CLUSTERING EFFECTS

**Number Of Clusters** Figure 5 shows how SPaRFT's performance depends on the number of clusters $K$. Accuracy peaks with a moderate number ($K = 7$), while very few ($K = 1$) or many ($K = 20$) clusters reduce performance. This reflects a trade-off between specialization and generalization: too few clusters collapse diverse examples into coarse groups, while too many fragment the data, causing sparse sampling and unstable training signals. A moderate clustering level provides the best balance, enabling the bandit to exploit informative variation without over-fragmentation. These results highlight the importance of tuning $K$ carefully, as it directly affects how well the curriculum leverages per-example information.

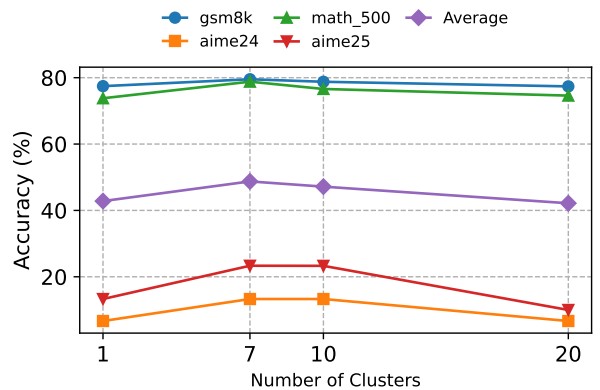

Figure 5: Results for 1 seed with Qwen3-0.6B and different number of clusters.

**Importance Of Concatenating Per-Example Attribute In Clustering** To assess the contribution of initial per-example attributes to cluster quality, we repeat the clustering pipeline after ablating this signal (i.e., omitting per-example attribute). Owing to space constraints, results are reported in Appendix Figure 8, showing downstream performance of *Qwen3-0.6B* on each dataset under this "no-attribute" condition. Across benchmarks, we observe a consistent drop in accuracy and reasoning metrics when per-example information is excluded, confirming that these attributes are key to forming meaningful clusters and improving curriculum selection.

## 6.4 OTHER ABLATION STUDIES

We also ablate (i) diverse sample selection, (ii) embedding model choice, (iii) number of PCA components, (iv) samples per cluster, (v) selected-sample difficulty, (vi) dataset distribution, and (vii) cluster properties and (vii) training time (Appendices A.5, A.6, A.7, A.9, A.10, A.11, A.12). Across these dimensions, the results consistently support the robustness of our approach, clarify trade-offs and hyperparameter sensitivities, and offer practical guidance for default settings.

## 7 CONCLUSION

We introduced **SPaRFT**, a lightweight framework that enables efficient reasoning in small language models through clustering and adaptive curriculum learning. SPaRFT selects compact, diverse training subsets and dynamically adapts training focus based on model performance. Experiments show that SPaRFT achieves competitive accuracy with significantly fewer samples. These results highlight the effectiveness of combining semantic clustering with performance-driven curricula to unlock reasoning in small models using minimal resources.

REPRODUCIBILITY STATEMENT

Implementation details and experimental setups are provided in the Appendix. Following publication, we will release our codebase as open source, along with documentation to facilitate reproducibility.

LLM USAGE

Large Language Models (LLMs) were not involved in the conception or design of our approach. We used them only to improve the manuscript's readability (grammar and style) and for a small post-hoc analysis; none of these uses influenced the method, training, or reported results.

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

# A APPENDIX

## A.1 ALGORITHM FOR SPARFT

In this section, we provide the pseudo-code for SPaRFT in Algorithm 1.

---

**Algorithm 1** SPaRFT

---

**Input**: Policy $\pi_\theta$, Dataset $\mathcal{D}$, Embedding model $\phi$, Clusters $K$, Batch size $B$, RL algorithm $\mathcal{A}$, Small positive constant $\epsilon$
**Output**: Trained policy $\pi_\theta$

1: **// Phase 1: Cluster-based Data Reduction**
2: **for** each $x_i \in \mathcal{D}$ **do**
3: $\quad$ $s_i \leftarrow \text{PCA}(\phi(x_i)), \quad d_i \leftarrow \text{difficulty}(x_i)$
4: $\quad$ $e_i \leftarrow s_i \oplus d_i$
5: **end for**
6: Run K-means on $\{e_i\}$ to form $\{C_k\}_{k=1}^K$
7: Select $l$ diverse examples $S_k \subset C_k$ via farthest sampling
8: $\mathcal{D}_{\text{train}} \leftarrow \bigcup_k \{x_i \mid i \in S_k\}$
9: **// Phase 2: Bandit-driven Curriculum**
10: Initialize Multi-arm Bandit
11: **for** each $k$ **do**
12: $\quad$ Initialize $R_k \leftarrow 0, n_k \leftarrow 0$
13: **end for**
14: **while** training not finished **do**
15: $\quad$ **for** each cluster $k = 1, \dots, K$ **do**
16: $\quad\quad$ Sample reward estimate: $\tilde{\mu}_k \sim \mathcal{N}\left(-\frac{R_k}{n_k+\epsilon}, \frac{1}{n_k+\epsilon}\right)$
17: $\quad$ **end for**
18: $\quad$ Select cluster: $c_t \leftarrow \arg\max_k \tilde{\mu}_k$
19: $\quad$ Sample batch $X \subset C_{c_t}$
20: $\quad$ Generate responses $G \leftarrow \pi_\theta(X)$
21: $\quad$ Calculate correctness $r_i, i = 1, \dots, B$
22: $\quad$ Compute average reward $r_{\text{avg}} \leftarrow \frac{1}{B} \sum_{i=1}^B r_i$
23: $\quad$ Update policy: $\pi_\theta \leftarrow \mathcal{A}(\pi_\theta, X, G, r_{\text{avg}})$
24: $\quad$ Update bandit stats:
$$R_{c_t} \leftarrow R_{c_t} + r_{\text{avg}}, \quad n_{c_t} \leftarrow n_{c_t} + 1$$
25: **end while**
26: **return** $\pi_\theta$

---

## A.2 CONVERGENCE OF THE THOMPSON SAMPLING SCHEDULER

We analyze the convergence of the Thompson Sampling scheduler used in SPaRFT. Each data cluster is treated as an arm in a multi-armed bandit. At step $t$, let $\pi_{\theta^{(t)}}$ denote the model with parameters $\theta^{(t)}$. The expected reward (solve rate) of cluster $\mathcal{C}_k$ is defined as:

$$\mu_k^{(t)} = \mathbb{E}_{x \sim \mathcal{C}_k} \left[ \Pr(\pi_{\theta^{(t)}}(x) = \text{ correct}) \right]. \tag{8}$$

where $\Pr\left(\pi_{\theta^{(t)}}(x) = \text{correct}\right)$ denotes the probability that the model produces a correct answer for input $x$.

We already have: (1) the model is trained using gradient clipping with threshold $G_{\max}$; (2) the learning rate $\alpha_t$ follows a cosine decay schedule with warmup and is therefore non-increasing and vanishes as $t \to \infty$; and (3) the expected rewards satisfy $\mu_k^{(t)} \in [0, 1]$ for all clusters $k$. These three properties are directly enforced in the SPaRFT implementation.

To complete the convergence analysis, we now bound the drift of each cluster's expected reward. Define

$$f_k(\theta) = \mathbb{E}_{x \sim \mathcal{C}_k}\left[\Pr\left(\pi_\theta(x) = \text{correct}\right)\right], \tag{9}$$

where $f_k$ is the *cluster-level reward surface* for arm $k$, and note that since each layer of our base model is continuously differentiable, so is $f_k(\theta)$ (Goodfellow et al., 2016). We assume that, in practice, gradient clipping at norm $G_{\max}$ together with a bounded initialization prevents the parameters $\{\theta^{(t)}\}$ from diverging excessively, effectively keeping them in some large but fixed ball $\{\|\theta\| \le R\}$. Empirical studies on LLM training have repeatedly observed that clipped updates under cosine-decay schedules yield stable trajectories without catastrophic parameter growth (e.g., (Wang et al., 2025a; Huang et al., 2025)).

Under this assumption, the extreme-value theorem (Rudin, 1976) guarantees the existence of a constant $H < \infty$ such that

$$\|\nabla_\theta f_k(\theta)\| \le H \quad \forall \, \|\theta\| \le R. \tag{10}$$

Moreover, each gradient step with cosine-decay learning rate $\alpha_t$ and gradient clipping satisfies

$$\|\theta^{(t+1)} - \theta^{(t)}\| \le \alpha_t \, G_{\max}. \tag{11}$$

Applying the mean-value theorem (Rudin, 1976) then yields

$$\begin{aligned}
\left|\mu_k^{(t+1)} - \mu_k^{(t)}\right| &= \left|f_k(\theta^{(t+1)}) - f_k(\theta^{(t)})\right| \\
&\le H \, \|\theta^{(t+1)} - \theta^{(t)}\| \\
&\le H \, G_{\max} \, \alpha_t \,=\, \varepsilon_t,
\end{aligned} \tag{12}$$

where $\varepsilon_t \to 0$ as $\alpha_t \to 0$. Thus, we obtain the desired vanishing drift $\left|\mu_k^{(t+1)} - \mu_k^{(t)}\right| \le \varepsilon_t$ for every cluster $k$.

Let $V_T = \sum_{t=1}^{T-1} \max_k |\mu_k^{(t+1)} - \mu_k^{(t)}|$ denote the total reward variation up to step $T$. The bound above implies

$$V_T \le \sum_{t=1}^{T-1} \varepsilon_t. \tag{13}$$

Since $\alpha_t = O(1/t)$ after warmup, we have $\sum_{t=1}^{T-1} \varepsilon_t = O(\log T)$, and hence $V_T = O(\log T)$. This sublinear variation is sufficient to ensure convergence of Thompson Sampling in the non-stationary bandit setting, as shown in prior work (Besbes et al., 2014). Therefore, the Thompson Sampling scheduler concentrates on the cluster(s) with the highest current expected reward as $t \to \infty$. This ensures that the bandit-based curriculum used in SPaRFT converges to the most informative training distribution over time.

### A.3 Additional Related Work: Reinforcement Fine-Tuning for Language Models

Due to page limit in the main paper, we include the additional related work in the Reinforcement Fine-Tuning methods for Language Models here.

Language Models can be formulated as sequential decision-making agents, enabling the application of RL techniques for fine-tuning. Proximal Policy Optimization (PPO) (Schulman et al., 2017) has been widely adopted in early RLHF pipelines due to its balance between stability and sample efficiency. More recent work introduced actor-only alternatives such as REINFORCE++ (Hu, 2025) and Group Relative Policy Optimization (GRPO) (DeepSeek-AI, 2025), which eliminate the need for value networks and have shown strong performance on large language models, particularly in reasoning tasks. By avoiding a separately trained critic, these approaches simplify optimization, reduce variance in policy updates, and mitigate instability caused by poorly estimated value functions. GRPO, in particular, has been successfully deployed in large-scale instruction tuning setups where explicit reward modeling is either impractical or misaligned with target behaviors. Instead of depending on hand-crafted reward models, GRPO leverages group-based relative comparisons across sampled trajectories, thereby aligning the optimization signal with preference-style supervision. This actor-only paradigm aligns naturally with recent trends in LLM alignment, where scalability, reduced computational overhead, and robustness to noisy feedback are critical. These methods represent a shift from critic-dependent RLHF pipelines toward lightweight, actor-centric algorithms that better match the scale and complexity of modern LLM training regimes.

## A.4 ADDITIONAL BASELINE: LEARNING IMPACT MEASUREMENT

In this section, we compare the performance of our method on mathematical reasoning tasks against a variance-based data selection approach, Learning Impact Measurement (LIM) (Li et al., 2025). While the original LIM paper reports results on the MATH-Full dataset (Hendrycks et al., 2021), we also conduct clustering on the same dataset using our method (SPaRFT) to ensure a fair comparison. It is worth noting that after data reduction, LIM retains approximately 1,400 training samples, whereas SPaRFT selects only 5 representative clusters, corresponding to just 50 data points—amounting to **merely 3% of the data used by LIM baseline**. Despite this drastic reduction, our method achieves competitive or superior results, demonstrating that SPaRFT can reach high efficiency and effectiveness with a fraction of the training data. For these experiments, we performs training on 3 different seeds, and report in Table 4.

### A.4.1 LIM DEFINITION

LIM computes a per-sample score from its reward trajectory relative to the model's average reward curve across epochs. Let $r_i^k$ be the reward of sample $i$ at epoch $k$ and $r_k = \frac{1}{N} \sum_{i=1}^{N} r_i^k$ the epoch-wise mean over all $N$ samples for $k = 1, \ldots, K$. The alignment score is

$$ s_i \;=\; 1 \;-\; \frac{\sum_{k=1}^{K} \left(r_i^k - r_k\right)^2}{\sum_{k=1}^{K} \left(1 - r_k\right)^2}, \qquad i = 1, \ldots, N, \tag{14} $$

which normalizes the squared deviation of the sample trajectory from the epoch-wise mean. Data reduction is performed by thresholding:

$$ \mathcal{D}_{\text{LIMR}} \;=\; \left\{\, i \,:\, s_i > \theta \,\right\}. \tag{15} $$

### A.4.2 RESULTS

We show the results on four mathematical reasoning datasets between our method and LIMR using MATH as training data in Table 4. We select Qwen2.5-0.5B-Instruct as the base LLM for training.

The results highlight the effectiveness of SPaRFT compared to both the Base model and LIMR. On GSM8K, SPaRFT reaches 32.0%, which represents a relative improvement of +22% over the Base (26.3%) and still surpasses LIMR (30.7%). On MATH500, SPaRFT achieves 20.7%, outperforming LIMR (20.3%) and the Base (20.0%), showing that our method yields more stable gains even on challenging competition-level problems. Notably, SPaRFT is the only method that improves performance on the Olympiad benchmarks: it attains 1.1% on AIME24 and 3.3% on AIME25, while both the Base and LIMR fail to make progress on these harder tasks.

Overall, these findings confirm that SPaRFT not only provides consistent improvements on standard benchmarks such as GSM8K and MATH500, but also uniquely enhances generalization to the most challenging settings, where variance-based selection methods like LIMR struggle. This demonstrates the robustness and efficiency of our cluster-based approach in leveraging limited training data for stronger downstream reasoning performance.

| Method | GSM8K | MATH500 | AIME24 | AIME25 |
|--------|-------|---------|--------|--------|
| Base | 26.3 | 20.0 | 0.0 | 0.0 |
| LIMR | $30.7 \pm 1.3$ | $20.3 \pm 1.1$ | $0.0 \pm 0.0$ | $0.0 \pm 0.0$ |
| SPaRFT | $\mathbf{32.0 \pm 0.8}$ | $\mathbf{20.7 \pm 1.2}$ | $\mathbf{1.1 \pm 1.9}$ | $\mathbf{3.3 \pm 0.0}$ |

Table 4: Mean $\pm$ standard deviation over 3 seeds on the MATH dataset using Qwen2.5-0.5B-Instruct as the base model. Best results are **bolded**.

## A.5 EFFECT OF DIVERSE SAMPLE SELECTION

We show the impact of sample selection strategies for the data reduction on the performance of different LLMs in Figure 6. Specifically, we compare our method against two baselines: *random*,

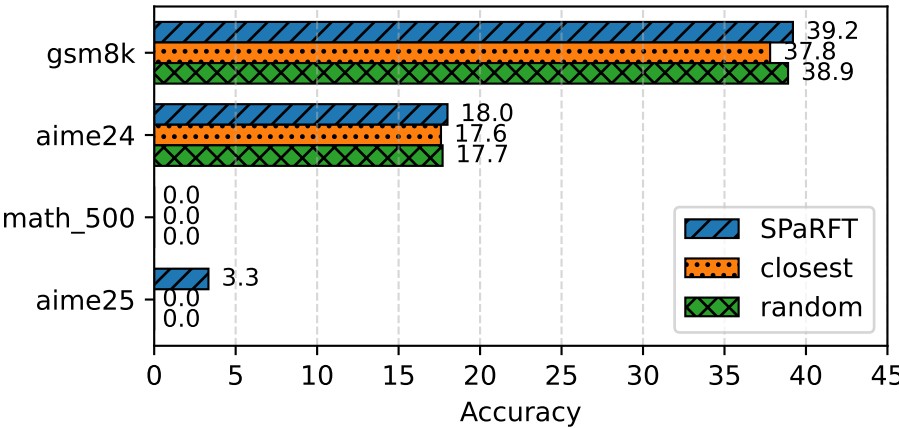

Figure 6: Comparison of selection strategies across datasets. Selecting diverse examples with SPaRFT outperforms both random and closest baselines. Closest examples perform worse, likely due to reduced variety within each cluster.

which randomly selects training examples for the cluster, and *closest*, which selects the closest examples to the cluster center. As observed, selecting diverse examples with our method consistently yields the highest performance across four datasets. Interestingly, the *closest* strategy performs worse than *random* in most cases. We hypothesize that this is because the examples nearest to the cluster center tend to be overly similar, thus failing to provide sufficient coverage and variation for effective learning.

## A.6 IMPACT OF EMBEDDING MODEL CHOICE

To assess SPaRFT's robustness to different semantic embedding backbones, we compare its default sentence encoder with an alternative based on *Qwen2-1.5B-Instruct* (*Alibaba-NLP/gte-Qwen2-1.5B-instruct*). We denote Qwen3[1] as the baseline using *Qwen3-Embedding-0.6B*, which is adopted in SPaRFT, and Qwen2.5[2] as the baseline using *Alibaba-NLP/gte-Qwen2-1.5B-instruct*. Table 5 reports zero-shot performance on four math reasoning benchmarks. Results show that SPaRFT yields nearly identical performance across both embedding models, with the Qwen3-based encoder showing a slight advantage. This plug-and-play flexibility demonstrates that any high-quality pre-trained encoder can be seamlessly integrated into our framework without retraining.

| Embedding | GSM8K | MATH500 | AIME24 | AIME25 |
|---|---|---|---|---|
| Qwen3[1] | 32.9±0.9 | 22.0±0.7 | 2.2±1.9 | 1.1±1.9 |
| Qwen2.5[2] | 31.6±0.6 | 21.0±1.2 | 1.1±1.9 | 0.0±0.0 |

Table 5: Results using DeepScaleR as training data for Qwen2.5-0.5B-Instruct, evaluated with two embedding models: Qwen3[1] and Qwen2.5[2].

## A.7 IMPACT OF THE NUMBER OF PCA COMPONENT

In SPaRFT, we first apply PCA to reduce the dimensionality of the latent vectors extracted from the pretrained Sentence-BERT model. By default we use 50 principal components; here, we vary this number to study its effect on final performance. Using Qwen3-0.6B as the base model and training on the DeepScaleR-uniform dataset, the results are shown in Table 6. We observe that smaller to moderate numbers of components (10 or 50) yield the best performance, whereas larger values (100 or 300) lead to a decline. We hypothesize that very high-dimensional embeddings overwhelm the difficulty signal prior to clustering, resulting in poorer downstream performance.

| Number of PCA Components | GSM8K | MATH500 | AIME24 | AIME25 | Average |
|---|---|---|---|---|---|
| 10 | 79.2 | 74.4 | 13.3 | 20.0 | 46.7 |
| 50 | 79.5 | 78.9 | 13.3 | 23.3 | 48.8 |
| 100 | 79.8 | 76.6 | 6.7 | 13.3 | 44.1 |
| 300 | 79.0 | 75.8 | 16.7 | 10.0 | 45.4 |

Table 6: Results on 1 same seed on the DeepScaleR-uniform dataset using Qwen3-0.6B as the base model with different number of PCA components.

## A.8 NUMBER OF SAMPLES IN EACH CLUSTER

We vary the number of samples per cluster $l$ to assess its impact, and present the results in Figure 7. As shown, performance peaks at our default setting of $l = 10$. In contrast, using $l = 1$ yields lower performance, likely because the selected examples lack sufficient diversity to provide a strong learning signal. Larger values of $l \in [100, 300]$ also lead to degraded performance, which we attribute to increased randomness when sampling too many examples per arm, especially when $l \gg B$. Due to compute limits, we couldn't extensively tune $l$ between 10–100, where better results might be possible. Overall, setting $l \sim B$ achieves the most reasonable results.

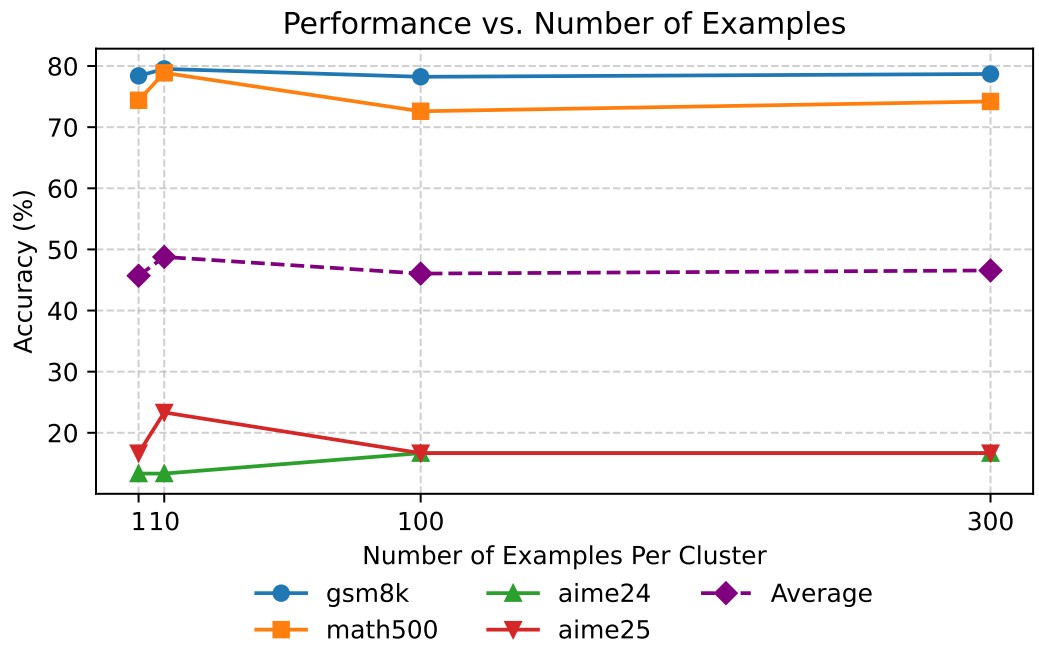

Figure 7: Results for 1 seed with Qwen3-0.6B and different number of samples per cluster.

## A.9 SELECTED TRAINING SAMPLES DIFFICULTY

We analyze the selected training examples to understand how SPaRFT constructs its curriculum. Figure 9 shows the difficulty distribution of questions selected on the DeepScaleR-uniform dataset. SPaRFT consistently chooses examples across the full difficulty range—from easy (near 0) to hard (near 100)—ensuring balanced coverage. This diversity enables training on a broad range of problems, avoiding overfitting to simple or complex cases. Notably, this balance emerges without manual difficulty constraints, highlighting the effectiveness of SPaRFT's clustering and selection strategy.

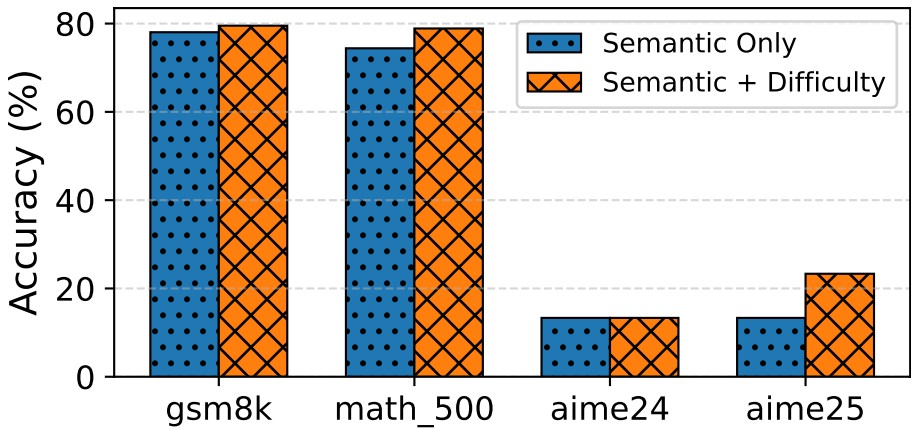

Figure 8: Effect of Removing Difficulty in Clustering on Performance.

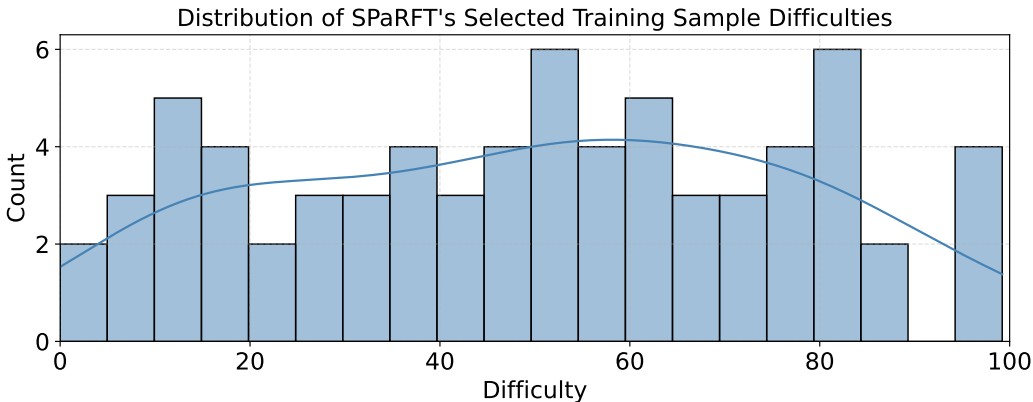

Figure 9: Difficulty distribution of training examples in SPaRFT.

## A.10   DEEPSCALER SUBSET DISTRIBUTIONS

We provide the difficulty score distributions of three DeepScaleR subsets: DeepScaleR Uniform, DeepScaleR Easy, and DeepScaleR Difficult, as shown in Figure 10. Each subset exhibits distinct difficulty characteristics, reflecting the varying levels of challenge present in the data. The distributions are grouped into bins of size 10, allowing for a clear comparison of how problem difficulty varies across these subsets. In particular, the Uniform subset spans the entire difficulty range with roughly balanced coverage, making it suitable for general-purpose training and evaluation. By contrast, the Easy subset is concentrated heavily in the lower-difficulty bins, highlighting its role in providing simpler problems for warm-up training or curriculum learning. Meanwhile, the Difficult subset skews strongly toward the higher-difficulty bins, offering more challenging samples that are valuable for stress-testing reasoning capabilities and benchmarking advanced methods. Together, these subsets provide complementary perspectives on model performance across a wide spectrum of difficulty levels, ensuring a more comprehensive assessment of reasoning ability.

## A.11   CLUTER ANALYSIS

In this section, to provide further insight into what happens during the clustering phase of our framework, we analyze several representative settings to examine both what is captured for training and how clustering shapes the overall behavior of our approach. Specifically, we investigate how problem embeddings are grouped and how these clusters align with meaningful curricular categories.

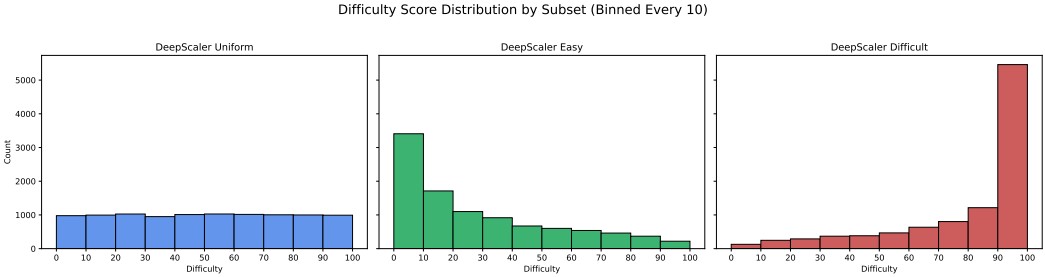

Figure 10: DeepScaleR subsets' difficulty distributions.

For this purpose, we employ GPT-5's API (OpenAI, 2025) to categorize each problem into one of the seven canonical subject areas defined by Hendrycks et al. (2021), namely: Prealgebra, Algebra, Number Theory, Counting & Probability, Geometry, Intermediate Algebra, and Precalculus. This taxonomy is consistent with the Art of Problem Solving (AoPS) curriculum (Art of Problem Solving), which provides a widely accepted structure for organizing mathematical problem-solving skills.

- **Prealgebra**: Covers arithmetic foundations, including fractions, decimals, percents, ratios, proportions, and basic number properties. It also introduces simple equations and word problems.

- **Algebra**: Focuses on symbolic manipulation and equations, such as linear and quadratic equations, inequalities, systems of equations, factoring, functions, and exponents. It marks the transition from arithmetic to general algebraic reasoning.

- **Number Theory**: Includes topics such as divisibility, prime numbers, greatest common divisors, modular arithmetic, congruences, and Diophantine equations. Problems emphasize reasoning about integer structure and properties.

- **Counting & Probability**: Encompasses combinatorics and elementary probability, including permutations, combinations, casework, binomial coefficients, expected value, and probabilistic reasoning.

- **Geometry**: Centers on Euclidean geometry of lines, angles, triangles, quadrilaterals, circles, and polygons. Topics include similarity, congruence, area, volume, coordinate geometry, and introductory trigonometric methods.

- **Intermediate Algebra**: Extends algebra with higher-level topics such as polynomials, rational functions, complex numbers, inequalities, logarithmic and exponential functions, and sequences/series.

- **Precalculus**: Prepares for calculus through trigonometry, advanced functions, polar/parametric representations, vectors, and deeper study of sequences and series.

### A.11.1 DEEPSCALER DATASET

We analyze the *DeepScaleR* dataset under the configuration with $K=7$ clusters induced by `Qwen3-0.6B-Embeddings`, consistent with Table 9. Figure 10 visualizes one representative run. The resulting partition exhibits intuitive curricular structure: a majority of items fall into *Prealgebra* (51.4%), with *Geometry* (18.6%) and *Algebra* (12.9%) also comprising substantial shares; by contrast, smaller categories such as *Number Theory* (1.4%) and *Intermediate Algebra* (1.4%) are comparatively scarce, with the remaining mass distributed across *Counting & Probability* and *Precalculus*.

Beyond mirroring topical prevalence in the underlying corpus, this distribution suggests that the embedding-driven clustering is aligned with both latent difficulty and high-level curricular distinctions. Practically, this yields two benefits for downstream training: (i) it avoids over-emphasizing any single subject area, providing balanced exposure across topics; and (ii) it simplifies scheduling, since strata can be sampled in a principled way (e.g., uniformly or by difficulty-aware policies) without ad-hoc reweighting to correct for cluster idiosyncrasies. In short, even though clusters are

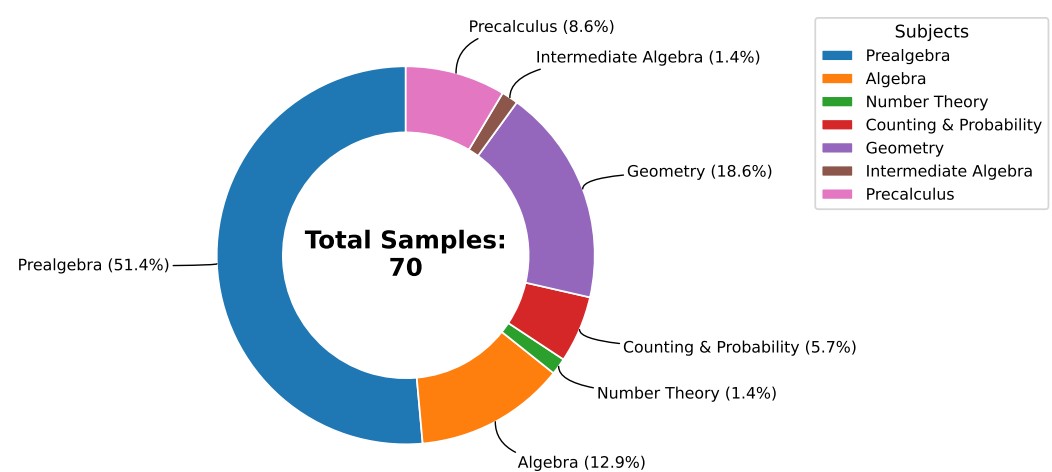

Figure 11: Selected data subjects (70 samples) using Qwen3-0.6B-Embedding. The data is clustered from DeepScaleR-uniform set.

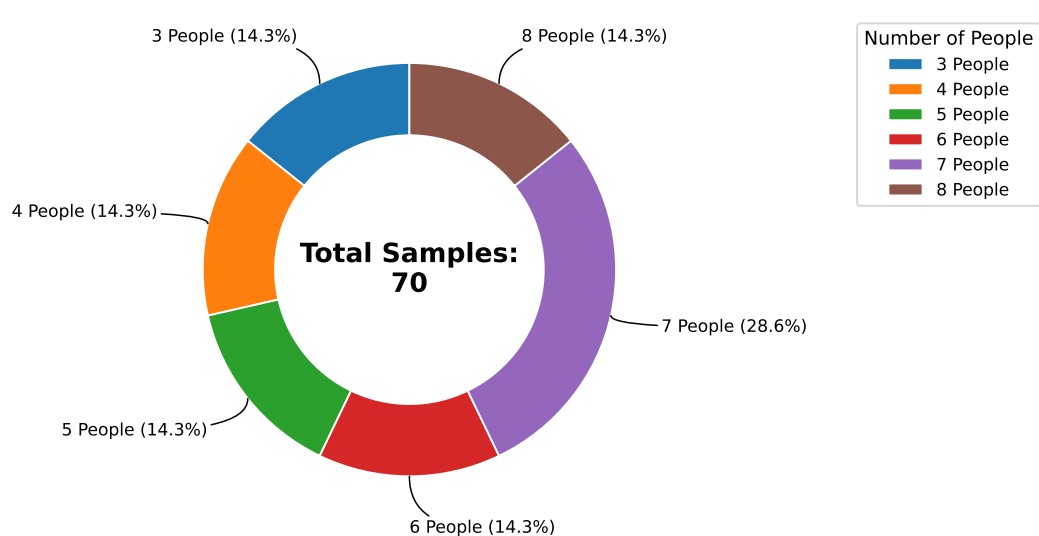

Figure 12: Selected data subjects (70 samples) using Qwen3-0.6B-Embedding. The data is clustered from Knights and Knaves set.

formed in representation space, they preserve pedagogically meaningful boundaries that support stable and fair curriculum design.

### A.11.2 KNIGHTS-AND-KNAVES DATASET

We conduct an analogous analysis on the *Knights-and-Knaves* dataset using $K=7$ clusters obtained with `Qwen3-0.6B-Embeddings`. The results are shown in Figure 12. Plotting the distribution of selected questions by the number of people per instance reveals two robust regularities across all three training seeds: (i) no 2-person questions are selected; and (ii) aside from the 7-person category, all remaining categories contain the same number of questions. The same symmetry appears in the other seeds, indicating that the clustering process is not only semantically coherent but also structurally consistent with a salient, coarse-grained attribute (the number of entities in the prompt).

These regularities have useful practical consequences. First, they provide balanced coverage over interaction sizes, preventing the curriculum from drifting toward a single conversational complexity.

Second, they reduce confounds in subsequent evaluation and scheduling: because most categories are equalized, one can adopt simple, uniform sampling or layer a performance-aware scheduler on top without introducing artifacts from cluster imbalance. We reckon the persistent absence of 2-person items likely reflects a combination of dataset composition and our selection protocol's preference for more discriminative examples.

## A.12 TRAINING TIME

To evaluate computational efficiency, we measure wall-clock training time when applying AdaRFT and our proposed SPaRFT across five representative base models: *Qwen3-0.6B*, *Falcon*, *Llama3*, *Qwen2.5*, and *Qwen3-8B*. These models span a range of parameter scales and architectures, providing a balanced testbed for assessing runtime behavior under different backbone choices. All runs are conducted under identical hardware and data conditions to ensure a fair comparison.

Figure 13 reports the results. Across all models, SPaRFT consistently reduces wall-clock training time relative to AdaRFT, with per-model savings ranging from $2.1\%$ to $10.8\%$ (e.g., Qwen2.5: $-66$ minutes, $-10.8\%$; Qwen3–8B: $-85$ minutes, $-5.6\%$; Llama3: $-40$ minutes, $-7.4\%$; Qwen3–0.6B: $-23$ minutes, $-2.4\%$; Falcon: $-14$ minutes, $-2.1\%$).

We attribute these savings to SPaRFT's clustering phase, which dynamically prioritizes clusters that yield higher learning signal over a reduced selection space, thereby avoiding wasted updates on redundant or low-yield samples. Although the absolute magnitude of savings depends on the underlying backbone, the improvements are consistent across diverse architectures, highlighting that SPaRFT not only improves data efficiency but also offers a practical reduction in training cost without additional engineering or inference-time overhead.

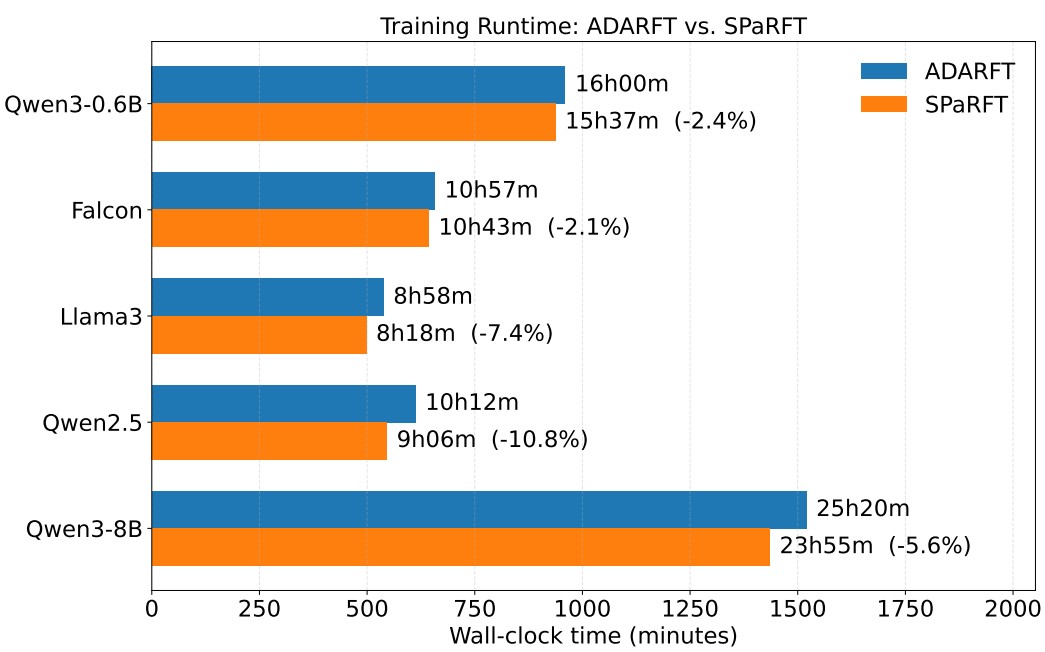

Figure 13: Training time comparison between AdaRFT and SPaRFT.

## A.13 SPaRFT TRAINING DETAILS

### A.13.1 TRAINING HYPERPARAMETERS

**General Training Parameters** In this section, we provide the training details of SPaRFT in Table 7. These parameters apply for full training LLMs (which excludes the training of *Qwen3-8B-Base*).

| Parameters | Value |
|---|---|
| Number of examples per cluster ($l$) | 10 |
| Number of PCA components | 50 |
| Batch size (B) | 8 |
| Number of generation per step (G) | 8 |
| Maximum completion length (L) | 1200 |
| Initial learning rate ($\alpha$) | $5e^{-6}$ |
| Weight Decay | 0.1 |
| Warmup Ratio | 0.1 |
| lr_scheduler_type | cosine |
| Adam $\beta_1$ | 0.9 |
| Adam $\beta_2$ | 0.99 |
| bf16 | True |
| Per device train batch size | 8 |
| Gradient accumulation steps | 8 |
| Max grad norm ($G_{norm}$) | 0.1 |
| $\epsilon$ | $1e^{-6}$ |

Table 7: Parameters used in SPaRFT.

**LoRA Training Parameters**    In this section, we provide the LoRA training parameters for *Qwen3-8B-Base*. All the parameters used are reported in Table 8.

| Parameters | Value |
|---|---|
| All Parameters | 8,194,569,216 |
| Trainable Parameters ($l$) | 3,833,856 |
| Trainable % | 0.05 |
| Rank | 8 |
| LoRA $\alpha$ | 16 |
| Target Modules | ["q_proj", "v_proj"] |
| LoRA Dropout | 0.1 |
| Bias | None |

Table 8: Parameters used for Low-rank Adaptation (LoRA) Fine-tuning.

A.13.2    NUMBER OF CLUSTERS

We provide the number of clusters used for different settings of SPaRFT in Table 9. While the optimal number of clusters varies across datasets, it remains within a moderate range, consistent with our observations in Section 6.3.

| Model | Train dataset | Number of clusters |
|---|---|---|
| Qwen3-0.6B | DeepScaleR-uniform | 7 |
| | DeepScaleR-easy | 8 |
| | DeepScaleR-difficult | 10 |
| | GSM8K | 10 |
| Falcon3-1B-Instruct | DeepScaleR-uniform | 6 |
| Llama-3.2-1B-Instruct | DeepScaleR-uniform | 8 |
| Qwen2.5-0.5B-Instruct | DeepScaleR-uniform | 7 |
| | Knights and Knaves | 7 |
| Qwen3-8B-Base | DeepScaleR-uniform | 7 |

Table 9: Number of clusters used for different settings of our method.

| Models/Datasets | URL |
|---|---|
| Qwen3-Embedding-0.6B | `https://huggingface.co/Qwen/Qwen3-Embedding-0.6B` |
| Qwen2.5-0.5B-Instruct | `https://huggingface.co/Qwen/Qwen2.5-0.5B-Instruct` |
| Llama3.2-1B-Instruct | `https://huggingface.co/meta-llama/Llama-3.2-1B-Instruct` |
| Falcon3-1B-Instruct | `https://huggingface.co/tiiuae/Falcon3-1B-Instruct` |
| Alibaba-NLP/ gte-Qwen2-1.5B-instruct | `https://huggingface.co/Alibaba-NLP/gte-Qwen2-1.5B-instruct` |
| DeepScaleR | `https://huggingface.co/datasets/agentica-org/DeepScaleR-Preview-` |
| GSM8K | `https://huggingface.co/datasets/openai/gsm8k` |
| MATH-500 | `https://huggingface.co/datasets/HuggingFaceH4/MATH-500` |
| AIME24 | `https://huggingface.co/datasets/math-ai/aime24` |
| AIME25 | `https://huggingface.co/datasets/math-ai/aime25` |

Table 10: Models and Datasets Details.

### A.13.3 MODEL AND DATA REFERENCES

We list the links to the LLM models and datasets in Table 10.

### A.13.4 SYSTEM PROMPT

Following (Hugging Face, 2025), the system prompt asks the model to generate the answer with clear requirements, with reasoning and answer following the format, as described in Figure 14.

> **SYSTEM PROMPT**
> You are a helpful assistant. Please reason step by step, and put your final answer within \boxed{}.

Figure 14: System prompt used in our experiments.

### A.14 RESPONSE EXAMPLES

We present several sampled responses of SPaRFT in Table 11 and 12.

### A.15 SELECTED EXAMPLE ANALYSIS

In this section, we provide details on the selected examples by SPaRFT. For each cluster, we show 2 selected examples selected by our method using *Qwen3-0.6B*, with the DeepScaleR-uniform dataset. We select the examples from the setting with total number of clusters equal 7. The examples are shown from Table 13 to Table 19.

Table 11: Qwen3-0.6B fine-tuned with SPaRFT responses.

---

**Question**

Let $f$ be the function defined by $f(x) = ax^2 - \sqrt{2}$ for some positive $a$. If $f(f(\sqrt{2})) = -\sqrt{2}$ then $a =$. Please reason step by step, and put your final answer within \boxed{}.

**Answer**

To solve for $a$, we need to find the value of $a$ such that $f(f(\sqrt{2})) = -\sqrt{2}$, where $f(x) = ax^2 - \sqrt{2}$. First, let's compute $f(\sqrt{2})$:

$$f(\sqrt{2}) = a(\sqrt{2})^2 - \sqrt{2} = a(2) - \sqrt{2} = 2a - \sqrt{2}$$

Now, we need to compute $f(f(\sqrt{2}))$:

$$f(f(\sqrt{2})) = f(2a - \sqrt{2}) = a(2a - \sqrt{2})^2 - \sqrt{2}$$

Let's expand the square term:

$$(2a - \sqrt{2})^2 = (2a)^2 - 2(2a)(\sqrt{2}) + (\sqrt{2})^2 = 4a^2 - 4a\sqrt{2} + 2$$

$$\cdots$$

So the discriminant is zero, which means there is a repeated root:

$$a = \frac{2\sqrt{2}}{4} = \frac{\sqrt{2}}{2}$$

Therefore, the value of $a$ is $\frac{\sqrt{2}}{2}$.

Let me double-check the calculations to ensure there are no errors. Starting from the beginning:

1. $f(\sqrt{2}) = a(2) - \sqrt{2} = 2a - \sqrt{2}$
2. Then $f(f(\sqrt{2})) = f(2a - \sqrt{2}) = a(2a - \sqrt{2})^2 - \sqrt{2}$
3. Expanding $(2a - \sqrt{2})^2$ gives $4a^2 - 4a\sqrt{2} + 2$
4. Multiplying by $a$ and subtracting $\sqrt{2}$ gives $4a^3 - 4a^2\sqrt{2} + 2a - \sqrt{2}$
5. Setting equal to $-\sqrt{2}$ leads to the equation $4a^3 - 4a^2\sqrt{2} + 2a = 0$
6. Factoring gives $2a(2a^2 - 2a\sqrt{2} + 1) = 0$, leading to the quadratic with discriminant zero.

Yes, the steps seem correct. The discriminant being zero implies a repeated root, which gives $a = \frac{\sqrt{2}}{2}$. Therefore, the answer is $\boxed{\dfrac{\sqrt{2}}{2}}$.

**Final Answer**

$$\boxed{\dfrac{\sqrt{2}}{2}}$$

---

Table 12: Qwen3-0.6B fine-tuned with SPaRFT responses.

**Question**
A set $S$ of points in the $xy$-plane is symmetric about the origin, both coordinate axes, and the line $y = x$. If $(2, 3)$ is in $S$, what is the smallest number of points in $S$? Please reason step by step, and put your final answer within \boxed{}.

**Answer**
To solve this problem, we need to determine the smallest number of points in the set $S$, given that the set is symmetric about the origin, both coordinate axes, and the line $y = x$. Additionally, the point $(2, 3)$ is in $S$.

—

### Step 1: Understand the Symmetry Conditions
A set of points that is symmetric about the **origin** means that for every point $(x, y)$ in $S$, $(-x, -y)$ is also in $S$.
Symmetry about the **coordinate axes** means that if $(x, y)$ is in $S$, then $(y, x)$ is also in $S$.
Symmetry about the **line $y = x$** means that if $(x, y)$ is in $S$, then $(y, x)$ is also in $S$.

$\cdots$

Is there a point that we might be missing?
Let's check if $(3, 2)$ is symmetric to itself. Yes, it is.
So, with these three points, we have covered all the symmetries.
But wait! What about the point $(2, 3)$? Is there a point symmetric to it that we haven't considered?
No, because we already considered all the points under the given symmetries.
So, the minimal number of points in $S$ is 3.

—

### Final Answer:

$$\boxed{3}$$

Table 13: Cluster 0 examples. Only 2 examples are shown for illustration.

**Example 1**
**Prompt**
(In the preliminaries of optimal method and experimental design) When using the 0.618 method to find the optimal amount to add in an experiment, if the current range of excellence is $[628, 774]$ and the good point is 718, then the value of the addition point for the current experiment is _____. Please reason step by step, and put your final answer within \boxed{}.
**Answer**
684

**Example 2**
**Prompt**
Calculate the probability that in a deck of 52 cards, the second card has a different suit than the first, and the third card has a different suit than the first and second. Please reason step by step, and put your final answer within \boxed{}.
**Answer**
$\frac{169}{425}$

Table 14: Cluster 1 examples. Only 2 examples are shown for illustration.

| |
|---|
| **Example 1** |
| **Prompt** |
| Calculate: $\frac{\cos 190°(1+\sqrt{3}\tan 10°)}{\sin 290°\sqrt{1-\cos 40°}} = $ _____. Please reason step by step, and put your final answer within \boxed{}. |
| **Answer** |
| $2\sqrt{2}$ |
| **Example 2** |
| **Prompt** |
| Let $a_n$ be the number of $n$-digit numbers formed using only digits 1 and 2 such that no two adjacent digits are both 2. Find $a_5$. Please reason step by step, and put your final answer within \boxed{}. |
| **Answer** |
| 13 |

Table 15: Cluster 2 examples. Only 2 examples are shown for illustration.

| |
|---|
| **Example 1** |
| **Prompt** |
| You have 5 red balls and 5 blue balls in a box. You randomly draw 4 balls without replacement. What is the probability that exactly 2 red balls are drawn? Please reason step by step, and put your final answer within \boxed{}. |
| **Answer** |
| $\frac{25}{63}$ |
| **Example 2** |
| **Prompt** |
| If a and b are real numbers such that $a^2 + b^2 = 1$, what is the maximum value of $ab$? Please reason step by step, and put your final answer within \boxed{}. |
| **Answer** |
| $\frac{1}{2}$ |

Table 16: Cluster 3 examples. Only 2 examples are shown for illustration.

| |
|---|
| **Example 1** |
| **Prompt** |
| Solve for $x$: $\log_3(x^2 - 1) = 2$. Please reason step by step, and put your final answer within \boxed{}. |
| **Answer** |
| 4 |
| **Example 2** |
| **Prompt** |
| Evaluate the integral $\int_0^1 xe^x \, dx$. Please reason step by step, and put your final answer within \boxed{}. |
| **Answer** |
| $e - 2$ |

Table 17: Cluster 4 examples. Only 2 examples are shown for illustration.

| |
|---|
| **Example 1** |
| **Prompt** |
| If $\sin x + \cos x = \sqrt{2}$, find the value of $\sin^4 x + \cos^4 x$. Please reason step by step, and put your final answer within \boxed{}. |
| **Answer** |
| $\frac{3}{4}$ |
| **Example 2** |
| **Prompt** |
| Find the sum of the series $\sum_{n=1}^{\infty} \frac{1}{n(n+1)}$. Please reason step by step, and put your final answer within \boxed{}. |
| **Answer** |
| 1 |

Table 18: Cluster 5 examples. Only 2 examples are shown for illustration.

| |
|---|
| **Example 1** |
| **Prompt** |
| How many 4-digit numbers are there such that no two adjacent digits are the same? Please reason step by step, and put your final answer within \boxed{}. |
| **Answer** |
| 5832 |
| **Example 2** |
| **Prompt** |
| If $A = \{1, 2, 3, 4\}$ and $B = \{3, 4, 5, 6\}$, what is $A \cup B$? Please reason step by step, and put your final answer within \boxed{}. |
| **Answer** |
| $\{1, 2, 3, 4, 5, 6\}$ |

Table 19: Cluster 6 examples. Only 2 examples are shown for illustration.

| |
|---|
| **Example 1** |
| **Prompt** |
| What is the value of the determinant of the matrix $\begin{bmatrix} 1 & 2 \\ 3 & 4 \end{bmatrix}$? Please reason step by step, and put your final answer within \boxed{}. |
| **Answer** |
| $-2$ |
| **Example 2** |
| **Prompt** |
| Simplify: $(2x - 3)^2 - (x + 1)^2$. Please reason step by step, and put your final answer within \boxed{}. |
| **Answer** |
| $3x^2 - 14x + 8$ |

