# OpenReview forum: "SPaRFT: Self-Paced Reinforcement Fine-Tuning for Large Language Models"
_ICLR.cc/2026/Conference — ICLR 2026 Conference Withdrawn Submission_

### Official Review · Reviewer_aDZN · 2025-10-30

**Soundness:** 3
**Presentation:** 2
**Contribution:** 2
**Rating:** 2
**Confidence:** 4

**Summary:**

The paper proposes SPaRFT, a method for improving data efficiency in Reinforcement Fine-Tuning for large language models. It combines clustering-based data reduction with Thompson-sampling-driven curriculum scheduling, where the agent focuses on more exploition on hard clusters and encourages explorations on clusters with fewer observations. The methods can reduce the number of training examples while introducing minimal computational overhead.

**Strengths:**

1. The paper addresses an important problem in LLM reinforcement fine-tuning. It has the potential to reduce the need for labeled data in RL with verifiable rewards.
2. It presents extensive experiments across multiple models and benchmarks, with thorough comparisons against relevant baselines.

**Weaknesses:**

1. There is a lack of justification for the reward signal design in Thompson sampling, where lower solve rates are prioritized. In GRPO, low-reward samples can have weak signals; this should be either empirically or theoretically justified.
3. There are some critical hyperparameters in this paper, but there are no well-justified methods to determine them, such as the number of PCA components, the number of cluster $K$, and the number of samples in each cluster. The paper mentioned it's important to the number of cluster K carefully and the number of samples in each cluster. However, it lacks a principled method for selecting these important hyperparameters. Empirically tuning with RL runs will be very expensive, which hinders the practicality of this method.
4. In terms of clustering features, both the representations of policy model and the difficulty labels can be model-dependent and evolve during training, so using static clusters might be an oversimplified choice.
5. Some figures and equations should be double checked, such as Equation (6) and Figure 6 (why the performances on math-500 are all zero?).

**Questions:**

1. Is the R1 and AdaRFT baseline trained on the full dataset or only on the 100 selected samples? It would be helpful to include a baseline trained on the selected dataset without the proposed curriculum (i.e., random selection) to isolate its effect.
2. Rather than only showing benchmark results, it will be good to report test performance across clusters to understand potential underfitting or overfitting to specific clusters and verify the theoretical results.
3. How are the difficulty labels generated or annotated? Is the clustering process sensitive to these labels, and how robust is it to noise or mislabeling in difficulty estimation?
4. How many total training steps are performed for each method? The paper should include training curve plots to visualize convergence behavior and compare learning stability across methods.

---

### Official Review · Reviewer_YC8t · 2025-10-31

**Soundness:** 2
**Presentation:** 2
**Contribution:** 2
**Rating:** 2
**Confidence:** 5

**Summary:**

The paper proposes a lightweight framework to improve the data efficiency of RFT for LLMs. It introduces a two-stage self-paced learning process: (1) cluster-based data reduction, which partitions training data by semantic and difficulty features to select compact, diverse subsets; and (2) bandit-based data assignment, which models each cluster as an arm in a multi-armed bandit and dynamically allocates samples according to current model performance. This adaptive curriculum prioritizes challenging yet learnable examples, reducing redundancy and training cost. Experiments on multiple reasoning benchmarks show that SPaRFT achieves comparable or higher accuracy than prior reinforcement and curriculum learning baselines while using up to 100× fewer training examples.

**Strengths:**

1. The paper addresses an important and timely problem in improving the data efficiency of reinforcement fine-tuning for LLMs.

2. The overall writing and structure are clear and easy to follow.

3. The paper reports standard deviations throughout most experiments, which reflects careful and transparent result reporting.

**Weaknesses:**

1. **Unclear clustering formulation and motivation.** The clustering step remains conceptually vague. Although the appendix provides some qualitative analysis, it is unclear why grouping examples by concatenating latent embeddings with difficulty scores should produce meaningful partitions for curriculum design. In particular, if difficulty already dominates the clustering signal, it is not obvious why the embedding feature is necessary. Conversely, if embeddings dominate, then similar-difficulty examples may be scattered across clusters, undermining the intended “self-paced” progression. The paper does not explain what specific characteristics each cluster is expected to capture or why a one-time, static clustering before training can remain valid as model capacity evolves. A deeper analysis (e.g., cluster interpretability, the evolution of cluster difficulty distribution over time, or comparison against purely difficulty-based grouping) would strengthen the method’s credibility.

2. **Experimental setup lacks rigor and completeness.** Most results are reported only on very small models (e.g., Qwen3-0.6B), which may not yield reliable or generalizable conclusions about RFT scaling. It is unclear whether all baselines are trained under identical sample budgets. Key implementation details, such as the number of clusters K, per-cluster sample count l, and batch size B, are missing from the main text and should be clearly stated to ensure reproducibility and fair comparison.

3. **Weak empirical support.** Many reported gains fall within the standard deviation range and are therefore statistically insignificant (e.g., Table 1 and Table 2). Figure 3 also offers limited evidence: on harder benchmarks such as AIME-24/25, the overall performance remains near zero, so the small numerical differences are uninformative. In some cases (e.g., MATH500), SPaRFT even underperforms the baselines. Table 3’s ablation likewise shows non-significant differences, suggesting that the claimed benefit of data reduction may be overstated. Overall, the empirical evidence remains weak.

**Questions:**

1. Please carefully address the concern raised in W1—I am particularly interested in the cluster difficulty distribution and its evolution over time. This analysis could clarify whether a one-time static clustering is sufficient or whether re-clustering during training would be more appropriate.

2. In line 256, what exactly does “only up to 100 training examples” mean? Please clarify whether this refers to l=100 (the number of samples per cluster) or the batch size B.

3. Related to W2, please specify whether the comparisons with baselines are fair—were all methods trained with the same total amount of data and compute?

4. For Qwen3-8B-Base, did you actually perform RFT? The mention of LoRA (Section 4) and the statement “not being used for RFT training” in line 376 seem confusing. Please clarify the setup.

5. (Minor) What is the intended takeaway from the right side of Figure 4? A more explicit interpretation would help readers understand its significance.

6. (Minor) Beyond the comparison with AdaRFT, I recommend discussing, and if possible comparing to, the recent online-data-selection works for LLM RFT.

[1] Wang, Q., Ke, J., Ye, H., Lin, Y., Fu, Y., Zhang, J., Keutzer, K., Xu, C. and Chen, Y., 2025. Angles Don't Lie: Unlocking Training-Efficient RL Through the Model's Own Signals. NeurIPS 2025.

[2] Sun, Y., Shen, J., Wang, Y., Chen, T., Wang, Z., Zhou, M. and Zhang, H., 2025. Improving Data Efficiency for LLM Reinforcement Fine-tuning Through Difficulty-targeted Online Data Selection and Rollout Replay. NeurIPS 2025.

---

### Official Review · Reviewer_jhDG · 2025-10-31

**Soundness:** 3
**Presentation:** 3
**Contribution:** 3
**Rating:** 6
**Confidence:** 3

**Summary:**

The paper proposes SPaRFT, a two‑stage framework for reinforcement fine‑tuning (RFT) of LLMs that (1) reduces training data via embedding+difficulty clustering with farthest‑point selection, and (2) assigns data online with a Thompson‑sampling multi‑armed bandit that favors clusters with lower current solve rates (harder for the model). On five reasoning benchmarks (GSM8K, MATH500, AIME24/25, Knights & Knaves), SPaRFT reportedly matches or exceeds curriculum and RL baselines while using up to 100× fewer unique training examples (≈100 picked) on small models, with small runtime overhead. Key results include gains on Qwen3‑0.6B and consistent improvements on other compact LLMs (Figure 3)

**Strengths:**

- Clear, modular method: The pipeline is simple and compatible with standard RFT (e.g., GRPO). The architecture diagram (Figure 1) makes the two‑phase design easy to follow, and Algorithm 1 in Appx A.1 spells out implementation details.

- Compelling data‑efficiency story: With ~100 selected examples, SPaRFT attains state‑of‑practice accuracies on GSM8K/MATH500/AIME24/25 (Table 1) and improves K&K (Table 2). The ablation that removes data reduction (Table 3) shows a measurable drop, supporting the necessity of Phase 1.


- Performance‑aware scheduling evidence: The bandit analysis heatmap (Figure 4‑Left) visualizes convergence toward moderately hard clusters; the difficulty‑over‑time plots (Figure 4‑Right) suggest SPaRFT avoids over‑sampling easy items compared to AdaRFT.

- Breadth of evaluations: Multiple datasets and model sizes (Qwen, Llama, Falcon; Figure 3) and sensible metrics (Lighteval extractive‑match; §4–5).

- Sensitivity studies: Effects of #clusters (Figure 5), samples/cluster (Figure 7), PCA components (Table 6), embedding backbones (Table 5), and selection strategy (Figure 6) provide practical guidance

**Weaknesses:**

1. Objective/sign convention vs. theory mismatch: The scheduler samples clusters with lower solve rates by negating the empirical mean in the Gaussian TS draw (Eq. 3) and selecting the argmax (Eq. 4). This implements hard‑first sampling. However, Proposition 1 then claims TS “concentrates on clusters with maximal expected reward” under sublinear variation (p. 5). Since “reward” earlier is the solve rate (Eq. 5–7), the proposition’s conclusion contradicts the negation actually used by the algorithm. Either the “reward” in the proposition should be redefinedor the statement should be inverted to “minimal expected reward.” As written, the theory does not match the implemented objective.


2. Thompson‑sampling modeling choice is ad‑hoc for Bernoulli rewards. nRewards are Bernoulli (correct/incorrect), yet the method samples from a Gaussian posterior with variance. No justification is given for the normal‑approximation/variance schedule (vs. a standard Beta–Bernoulli TS) nor for the effect of the negation on posterior updates. This creates ambiguity about the scheduler’s statistical correctness.

3. “100× fewer samples” and compute parity are under‑specified. The paper emphasizes using 100 unique examples but keeps training time nearly identical to R1 (15h37m vs 15h23m on Qwen3‑0.6B; §4) and shows only modest time savings versus AdaRFT (2–11%; Figure 13). It appears SPaRFT reduces unique training items but not the number of optimization steps. This weakens the “minimal resources” narrative and invites a stronger compute‑parity study (equal steps, equal tokens, and equal number of unique items for baselines).

4. Difficulty labels and preprocessing cost not accounted for. The data pipeline requires Qwen3‑Embedding‑0.6B encodings, PCA, k‑means, and a per‑example difficulty attribute (“solve rates from a moderate LLM” or explicit labels; §4). The wall‑clock comparisons seem to exclude these offline costs. Please quantify the one‑off preprocessing overhead and any cost of generating the difficulty signal.

5. Fairness and contamination controls. It is unclear whether train/test de‑duplication across DeepScaleR and evaluation datasets was performed. The claim that Qwen3‑0.6B becomes “comparable to larger models” lacks a cited head‑to‑head comparison under identical evaluation setups. Provide dedup checks and a precise comparison. (Table 1, §4–5.)


6. Statistical reporting. Many headline gains are small (e.g., GSM8K +0.6–1.6 points). While std‑devs over 3 seeds are reported (Table 1), please add confidence intervals and paired significance (especially for AIME24/25 and K&K Table 2, where absolute accuracies are low and variances non‑negligible).

**Questions:**

1. Theory alignment: Do you intend the bandit to optimize hardness (low solve rate) or reward (high solve rate)? Please restate Proposition 1 with the exact pseudo‑reward your TS uses (including the negation) and justify the Gaussian posterior choice for Bernoulli rewards (why not Beta–Bernoulli TS, or a GLM‑TS variant?). (§3.2; Appx A.2.)

2. Compute parity: For Table 1/2, how many optimizer steps, tokens, and unique examples did each baseline see? Can you add a matched‑budget comparison where SFT/R1/AdaRFT train on the same 100 unique items (random or heuristic selection) to isolate the effect of your scheduler and reduction? (§4–5.)

3. Contamination controls: What de‑duplication was performed between DeepScaleR and evaluation sets (GSM8K/MATH/AIME/K&K)? If none, can you provide a quick hash‑based de‑dup analysis? (§4–5.)

## Suggestion

- Fix theory/notation to align the negated mean with the proposition; consider reporting results for a Beta–Bernoulli TS varian
- Preprocessing cost: Report one‑off costs for embeddings, clustering, and difficulty estimation; clarify whether those costs are amortized across runs. (§3–4.)
- Statistical testing: Add 95% CIs and paired tests on Table 1/2; show seed‑wise scatter plots.
- Robustness: Include dedup checks, and probe performance when the per‑example attribute is noisy or absent (you partially ablate this in Figure 8; extend to a no‑difficulty pipeline with only embeddings).

---

### Official Review · Reviewer_pVem · 2025-11-03

**Soundness:** 2
**Presentation:** 2
**Contribution:** 2
**Rating:** 2
**Confidence:** 4

**Summary:**

The paper targets the problem of online data selection for reasoning tasks trained with GRPO. The authors first compute question embeddings concatenated with difficulty scores and apply PCA to cluster the questions. From each cluster, they select the farthest $l$ samples. During training, each cluster is treated as a bandit, from which the algorithm adaptively selects questions to draw. Experimental results show that the proposed data selection strategy outperforms baseline methods while using fewer samples.

**Strengths:**

The paper addresses an important challenge in reinforcement learning for large language models under sparse-reward settings. Incorporating an online data selection mechanism via a bandit framework is a promising direction to improve RL training efficiency and stability. The method is conceptually simple and practically implementable.

**Weaknesses:**

The primary weakness lies in the limited experimental scale. The proposed idea is only validated on small datasets and relatively small models. Although the method achieves performance gains in the limited-data regime, it remains unclear how well it generalizes to larger datasets and more capable models. Figure 7 seems to suggest that scaling data does not further improve performance gains, which raises questions about scalability and robustness.

**Questions:**

1. In Eq. (3), the bandit mean is defined on the negative mean reward, which implies that the hardest cluster will have a higher probability of being selected. Consider a worst-case scenario where the hardest cluster contains overly difficult questions such that all generations yield a reward of zero — in that case, there would be no gradient signal for GRPO, and the bandit mean for that cluster would remain zero, yet it would continue to be selected frequently. Would it make more sense to prioritize clusters whose cumulative reward is closest to 0.5 instead of purely minimizing it?
2. You use Qwen3-Embedding-0.6B to obtain question embeddings. Should the embedding model ideally belong to the same family as the base model used for RL training? If not, what is the potential impact of using a different embedding model? It might be more consistent to use the base model itself to derive the clustering representation.
3. When performing PCA, the latent representation is typically high-dimensional, while the difficulty score is one-dimensional. This dimensionality mismatch might make it difficult for PCA to fully leverage the difficulty signal. As observed in Figure 8, the inclusion of difficulty only slightly changes the outcome (notably in AIME25). Moreover, in Figure 4, the difficulty metric increases from 49.6 to 50.2, which seems marginal. In contrast, for your baseline ADARFT, the difficulty increases from less than 3 to about 6.5, which seems more meaningful.
4. Regarding difficulty scaling, RL training on mixed datasets often involves data sources with different difficulty scales. How would you handle combining datasets with inconsistent difficulty distributions or scales?
5. In Figure 4 (heatmap), cluster 3 is always selected exactly 70 times across different training steps. Should we not expect at least some randomness here? The paper explains: “In contrast, cluster 3, which is rarely picked, starts high at 70% and remains flat, suggesting it is too easy to help the model improve.” However, if this cluster is indeed too easy, shouldn’t the bandit algorithm naturally reduce its selection probability rather than keeping it constant?

---

### Note · Authors · 2025-11-24

I have read and agree with the venue's withdrawal policy on behalf of myself and my co-authors.